# Current Progress in the Development of Zika Virus Vaccines

**DOI:** 10.3390/vaccines9091004

**Published:** 2021-09-09

**Authors:** Kehui Zhou, Chaoqun Li, Wen Shi, Xiaodan Hu, Kutty Selva Nandakumar, Shibo Jiang, Naru Zhang

**Affiliations:** 1Department of Clinical Medicine, School of Medicine, Zhejiang University City College, Hangzhou 310015, China; 31704257@stu.zucc.edu.cn (K.Z.); 31704246@stu.zucc.edu.cn (C.L.); 31704195@stu.zucc.edu.cn (W.S.); 31704244@stu.zucc.edu.cn (X.H.); 2School of Pharmaceutical Sciences, Southern Medical University, Guangzhou 510515, China; 3Key Laboratory of Medical Molecular Virology (MOE/NHC/CAMS), School of Basic Medical Sciences, Shanghai Institute of Infectious Disease and Biosecurity, Fudan University, Shanghai 200032, China

**Keywords:** Zika virus, vaccine, different strategies

## Abstract

Zika virus (ZIKV) is an arbovirus first discovered in the Americas. ZIKV infection is insidious based on its mild clinical symptoms observed after infection. In Brazil, after 2015, ZIKV infection broke out on a large scale, and many infected pregnant women gave birth to babies with microcephaly. The teratogenic effects of the virus on the fetus and its effects on nerves and the immune system have attracted great attention. Currently, no specific prophylactics or therapeutics are clinically available to treat ZIKV infection. Development of a safe and effective vaccine is essential to prevent the rise of any potential pandemic. In this review, we summarize the latest research on Zika vaccine development based on different strategies, including DNA vaccines, subunit vaccines, live-attenuated vaccines, virus-vector-based vaccines, inactivated vaccines, virus-like particles (VLPs), mRNA-based vaccines, and others. We anticipate that this review will facilitate further progress toward the development of effective and safe vaccines against ZIKV infection.

## 1. Introduction

Zika virus (ZIKV) is a small envelope, positive-strand RNA virus belonging to the Flavivirus family of *Flaviviridae* [1]. As shown in Figure 1, the genome-encoded polyprotein can be cleaved into three structural proteins (capsid (C), anterior membrane (prM), and envelope (E)) and seven non-structural proteins (NS1, NS2A, NS2B, NS3, NS4A, NS4B, and NS5) [2]. The mature ZIKV particles consist of 90 E homodimers and 90 M homodimers on the lipid membrane, and the genomic RNA is surrounded by C protein (Figure 2). E proteins are responsible for receptor binding, attachment, viral entry, and membrane fusion. Transmission routes are arthropod vectors (e.g., *Aedes aegypti*), intrauterine (perinatal), and through sex and blood-related pathways [3]. ZIKV was also found to be present in the breast milk [4,5]. When ZIKV attacks pregnant women, it is easily reproduced in the placental tissue and seriously affects the fetal central nervous system and immune system, causing congenital Zika syndrome (CZS) in infants [6,7,8]. Studies have identified the major epitopes present on ZIKV structural proteins that can induce neutralizing antibodies [9]. Owing to high structural homology, the presence of common epitopes between Dengue virus (DENV) and ZIKV were reported earlier [10]. However, it is not feasible to prepare a ZIKV vaccine based on the principle of “cross-reactivity of the neutralizing antibodies” [11]. Several in vitro experiments demonstrated an antibody-dependent enhancement (ADE) of ZIKV infection after DENV infection, which poses a challenge to the development of a safe vaccine. In this review, we systematically discuss different types of Zika vaccines currently under development.

## 2. Development of Vaccines against ZIKV Infection

Development of a safe and an effective vaccine plays an important role in preventing the potential spread and serious harm caused by of ZIKV infection. Here we summarize and discuss different kinds of vaccines against ZIKV infection, including DNA vaccines, subunit vaccines, live-attenuated vaccines, virus-vector-based vaccines, inactivated vaccines, virus-like particle (VLP)- and mRNA-based vaccines.

### 2.1. DNA Vaccines against ZIKV Infection

The DNA vaccine platform has been used for over twenty-five years to develop candidate vaccines against numerous pathogens. DNA vaccines can induce both humoral and cellular immune responses and are capable of mediating long-term protection [12]. Most currently developed DNA vaccines for ZIKV contain prM and E genes coding for prM and E proteins (Table 1). Type I interferon receptor alpha-chain null mice (*Ifnar1*^−/−^ mouse model) [13,14] exposed to ZIKV developed severe damage to the testes and sperm [15,16], but a DNA vaccine encoding ZIKV prM-E completely protected mice against such ZIKV-associated damage [17]. The immunogenicity of a DNA-based vaccine candidate, pVAX1-ZME, expressing the prM/E protein of ZIKV, was evaluated in maternal and post-natal protection of suckling BALB/c mice, and it was demonstrated that the administration of three doses with 50 µg of pVAX1-ZME by in vivo electroporation induced robust ZIKV-specific cellular and long-term humoral immune responses with high and sustained neutralizing activity in adult BALB/c mice. The neutralizing antibodies passively protected against ZIKV infection in neonatal mice and effectively inhibited delay in growth [18]. GLS-5700, a DNA-based vaccine that encodes the prM and E antigenic regions of ZIKV, was shown to prevent fertility loss in male IFNAR^−/−^ mice [19]. The Vaccine Research Center (VRC) of the National Institute of Allergy and Infectious Diseases (NIAID) and National Institutes of Health (NIH) in USA have developed two DNA vaccine candidates, named VRC5288 and VRC5283, and tested them in phase I clinical trials to assess their safety, tolerability, and immunogenicity in humans [20]. Another DNA vaccine was developed by using a single tetrafunctional amphiphilic block copolymer (ABC) encoding the full sequence of prM-E, which induced a high neutralizing antibody titer against three divergent ZIKV isolates in six-week-old female C57BL/6C mice [21].

Three different ZIKV NS1 DNA vaccines (encoding wild-type NS1 (pVAX-NS1), secreted NS1 with a tissue plasminogen activator (TPA) leader sequence introduced upstream of the NS1 to ensure efficient secretion (pVAX-tpaNS1), or NS1 secreted as a heptamer by fusing to a chimeric version of the oligomerization domain from the chicken complement inhibitor C4b-binding protein, termed as IMX313P (pVAX-tpaNS1-IMX313P)) were evaluated in 6- to 8-week-old BALB/c and IFNAR^−/−^ mice. Results showed that the NS1-specific antibody titers and CD4^+^, as well as CD8^+^, T-cell responses induced by pVAX-tpaNS1 vaccination were significantly higher and stronger than those induced by pVAX-NS1 and pVAX-tpaNS1- IMX313P [22]. This study highlights the importance of NS1 as a target for protective Zika vaccines and reaffirms the notion that TPA-driven NS1 secretion determines the immunogenicity of ZIKV NS1 in a DNA vaccine. Such an NS1 DNA vaccine might offer an attractive alternative to envelope-based vaccines because DNA vaccines targeting the NS1 gene do not have the risk of inducing ADE in individuals living in areas endemic for DENV and other flaviviruses. 

DNA vaccines offer optimization of the sequences of the encoding genes in a flexible manner and the ability to test multiple candidate antigens rapidly [23]. They are chemically stable and easy to produce, with no requirement for cold-chain storage, and they are also cost-effective to manufacture on a large scale [22,24]. However, DNA vaccines involve expression of multiple foreign genes and thus have the potential of integrating the exogenous gene into the host genome, leading to induction of host autoimmunity [25].

### 2.2. Subunit Vaccines against ZIKV Infection

Based on their rapid, stable, and consistent production capabilities, subunit vaccines are considered as effective tools to prevent virus infection. Subunit vaccines against ZIKV have been designed and tested in animal models (Table 2). Viral structural proteins, such as E protein and its domain III (EDIII), are attractive vaccine targets. The immunogenicity of a subunit vaccine candidate comprised of ZIKV E protein and two clinical adjuvants (Alum and CoVaccine HT^TM^) was evaluated in Swiss Webster, BALB/c, and C57BL/6 mice [26]. These vaccine formulations induced robust IgG titers and high levels of neutralizing antibodies in all three mouse strains and protected them against viremia after ZIKV infection [26]. Another research team used a recombinant subunit platform consisting of antigens produced in *Drosophila melanogaster* S2 cells to develop two candidate formulations. The first formulation contained 25 µg of ZIKV E which was adjuvanted with 10 mg Co-Vaccine HT^TM^, and the second formulation contained 50 µg of ZIKV E protein with Alhydrogel^®^ 85 plus 1.2 mg of elemental aluminum. High neutralizing antibody titers were induced in a non-human primate (NHP) viremia model, and passive transfer of the plasma from the macaques protected against viremia in ZIKV-infected BALB/c mice [27]. Based on this observation, another research team agreed that high anti-ZIKV titers protected against viremia, but they also suggested that low titers could provide an incremental degree of protection, albeit not sufficient to prevent viremia [26]. Purified EDIII from transformed *Escherichia coli* inclusions induced high titers of IgG and ZIKV neutralizing antibodies, which showed no evidence of ADE induction in C57BL/6 mice [28]. 

A truncated subunit vaccine consisting of the first 450 amino acids at the N-terminal region of the ZIKV FSS13025 strain E protein (E90) was investigated in 7- to 8-week-old CD-1 (ICR) immunocompetent mice for both in utero and neonatal ZIKV infection. Results demonstrated that immunization of pregnant mice with E90 protected the developing brains of offspring, both in utero and during the neonatal period, from subsequent ZIKV infection and microcephaly. Most importantly, E90 vaccination protected mice from ZIKV infection, even at 140 days post-immunization [29]. Another group showed robust induction of ZIKV-specific humoral response in adult BALB/c mice by E90, and passive transfer of the antisera from these mice conferred absolute protection against lethal ZIKV challenge in a neonatal mouse model [30]. These studies demonstrated the promising nature of recombinant ZIKV E90 as a ZIKV subunit vaccine that deserves further clinical development. One study used seven-day-old male and female BALB/c pups, 6 to 8-week-old female BALB/c mice, and 5-week-old male and female type-I IFN receptor–deficient A129 mice to investigate the long-term immunogenicity and neutralizing activity of the ZIKV EDIII fragments. The results showed that ZIKV EDIII fragment, especially E298–409, could induce sustained development of neutralizing antibodies [31]. The E298–409-specific antibodies upon passive transfer prevented ZIKV infection in newborns and immunocompromised adults [31]. Thus, this subunit vaccine based on the critical fragment (E298–409) of ZIKV EDIII is one of the promising vaccine candidates for ZIKV infection. It has been proven that the recombinant ZIKV subunit vaccine is a safe and efficacious option for the prevention of ZIKV infection. However, its less immunogenic nature is the major disadvantage, requiring more doses and appropriate adjuvants [32].

### 2.3. Live-Attenuated Vaccines against ZIKV Infection

Live-attenuated vaccines (LAV) are those that reduce the infectivity of pathogens after various treatments, but retain their immunogenicity. A number of live-attenuated vaccines have been evaluated in animal models (Table 3). Candidate LAV viruses with engineered deletions in the 3′ untranslated region (UTR) provide immunity and protection in animal models of ZIKV infection [33]. Recently, a LAV candidate containing a 10-nucleotide deletion in the 3′ UTR of the ZIKV genome (10-del ZIKV) was developed. Even when immunized at a low dose with only 10-del ZIKV, complete protection from viremia by the induction of a high level of neutralizing antibodies was observed, preventing a decrease in the sperm count in A129 mice [34]. Another LAV candidate containing a 20-nucleotide deletion in the 3′ UTR of the ZIKV genome prevented viral transmission during pregnancy and testis damage in mice, as well as infection of NHP [35]. The ZIKV-3′UTR-Δ20-LAV is less sensitive to type-I-interferon inhibition than ZIKV-3′UTR-Δ10-LAV [34], making it more virulent than ZIKV-3′ UTR-Δ10-LAV in A129 mice. Collectively, LAV candidates containing deletions in the 3′ UTR of the ZIKV genome are efficacious and have the potential to be promising vaccine candidates. 

A live-attenuated ZIKV vaccine candidate encoding NS1 protein, but without glycosylation (ZIKV-NS1-LAV), was demonstrated to markedly diminish viral RNA levels in maternal, placental, and fetal tissues, which resulted in protection against placental damage and fetal death [36]. A single-dose LAV candidate containing a 9-amino-acid deletion in the viral capsid protein that infects cells with controlled, limited infection rounds was developed to test its safety and immunogenicity in A129 mice [37]. The results showed that a single-dose immunization of this LAV vaccine elicited protective immunity that completely prevented viremia, morbidity, and mortality. At the same time, it also fully prevented infection of pregnant mice and maternal-to-fetal transmission. Interestingly, injection of this vaccine with 10^4^ plaque-forming units to 1-day-old mice did not cause any disease or death, underscoring its safety [37].

The need for only a single dose, rapid onset of immune responses, and durable protection are the advantages of LAVs [24], but manufacture and transport of LAVs require cell-culture and cold-chain storage facilities, which are not often feasible in many countries. Combining the strengths of LAV and DNA vaccines may overcome these limitations. In this direction, a DNA-LAV was developed by engineering the cDNA copy of a ZIKV LAV genome into a DNA plasmid administered into A129 mice through the intramuscular route by using a clinically proven device, TriGrid^TM^, to initiate the replication of LAVs [24]. A single-dose immunization with 0.5 µg of the DNA-LAV vaccine elicited robust T-cell responses and production of high levels of neutralizing antibodies, which completely prevented testis infection, injury, and oligospermia in male mice [24]. This DNA-LAV vaccination also fully protected against ZIKV-induced disease and maternal-to-fetal transmission in pregnant mice [24]. Thus, the DNA-LAV approach is a promising platform for developing effective vaccines against ZIKV infection.

### 2.4. Virus-Vector-Based Vaccines against ZIKV Infection

Virus-vector-based vaccines of ZIKV are designed to introduce ZIKV genetic material into cells using a virus as the carrier to induce protective immunity and achieve lasting protection (Table 4). Lentivirus, retrovirus, and adeno-associated virus can be used as carriers. Adenovirus vectors (AAV) can reduce the risk of insertion mutagenesis, induce strong innate immune and adaptive immune responses in mammalian hosts, and are easy to use for genetic modifications [38]. Thus, AAVs are widely used in the treatment of infectious diseases. Adenovirus-vectored vaccines represent a favorable controlling strategy for the ZIKV epidemic. Two adenovirus-vectored Zika vaccines were constructed by inserting a ZIKV prM-E gene expression cassette into human adenovirus type 4 (Ad4-prM-E) and 5 (Ad5-prM-E) vectors [39]. A replication-defective vector which contains the full length of prM-E genes of ZIKV PRVABC59 strain [39] was created. Surprisingly, ELISA and plaque reduction neutralization tests showed negligible levels of anti-ZIKV antibodies after Ad4-prM-E-vaccination in C57BL/6 mice, suggesting that Ad4-prM-E vaccination induces only T-cell responses, whereas Ad5-prM-E vaccination induced both anti-ZIKV antibody and T-cell response [39]. Interestingly, coadministration of UV-inactivated Ad4 vector with Ad5-prM-E vaccine led to significant reduction in CTL and overall T-cell responses, compared to Ad5-prM-E alone [40], highlighting the differences in serotype-specific immunity induced by adenovirus vectors.

An intranasal Zika vaccine based on the replication-deficient hAd5 expressing ZIKV prM and E proteins recognized by the human antibody repertoire was able to induce both cell-mediated and humoral immune responses, which conferred protection against ZIKV challenge as demonstrated in a preclinical model of ZIKV infection [41]. Two novel hAd5-vector vaccines expressing ZIKV prM-E (Ad5-Sig-prM-Env) and E (Ad5-Env) proteins were constructed and evaluated in multiple non-lethal and lethal animal models using epidemic ZIKV strains [42]. Both vaccines elicited robust humoral and cellular immune responses in immunocompetent BALB/c and A129 mice, but Ad5-Sig-prM-Env-vaccinated mice had significantly higher ZIKV-specific neutralizing antibody titers and lower viral loads than Ad5-Env-vaccinated mice, suggesting that the Ad5-Sig-prM-Env vaccine was more immunogenic [42]. Four replication-deficient chimpanzee adenoviral (ChAdOx1) ZIKV vaccine candidates having addition or deletion of prM and E, with or without its transmembrane domain (TM), were designed and evaluated in vivo [43]. A single dose of ChAdOx1 ZIKV vaccine, without adjuvant, induced protective responses in ZIKV-challenged BALB/c mice [43]. ChAdOx1 prME, encoding prM, but with TM deletion, conferred 100% protection with long-lasting anti-E immune response and no ADE to dengue virus infection [43]. On the other hand, deletion of prM and addition of TM reduced protective efficacy and yielded lower anti-E responses [43]. This study highlights the importance of rational design of viral-vectored ZIKV vaccines for best protective responses. A single-slot recombinant rhesus adenovirus serotype 52 (RhAd52) vector-based vaccine, expressing ZIKV prM and E, induced ZIKV-specific neutralizing antibody responses, as well as E-specific cellular immune responses in rhesus monkeys, conferring complete protection against ZIKV challenge [44]. The genetic structure of rhesus monkey adenoviruses resembles that of other human or chimpanzee adenoviruses; therefore, rhesus adenovirus vectors have the potential advantage to be used as a novel class of vaccine vectors [47]. An attenuated recombinant vesicular stomatitis virus (rVSV)-based vaccine expressing ZIKV prM-E-NS1 as a polyprotein was tested in BALB/c and A129 mouse models. This vector vaccine induced ZIKV-specific antibodies and T-cell responses that conferred partial protection against ZIKV infection [45]. In order to avoid ADE-related effects, a recombinant VSV-based vaccine carrying the ZIKV strain PRVABC59 capsid protein (VSV-Capsid) was generated, and the recombinant VSV-ZikaE260-425 virus expressing amino acids 260-425 of ZIKV EDIII was used as a parallel control [46]. Both VSV-Capsid and VSV-ZikaE260-425 vaccines induced strong ZIKV-specific humoral responses in immunized BALB/c mice, but VSV-Capsid immunization elicited significantly higher levels of IFN-γ+ CD8^+^ and CD4^+^ T-cells compared to the VSV-ZikaE260-425 vaccine, demonstrating that the VSV-Capsid vaccine conferred more effective protection upon ZIKV challenge [46]. This study provided insights into the importance of ZIKV capsid protein for further development of Zika vaccines.

The virus-vector-based Zika vaccines have high efficiency in inducing a faster immune response. However, some vectors have pre-existing immunity in human, which significantly prevents their use in the development of promising vaccine candidates. For example, humans have neutralizing antibodies against AAV vectors [48,49] that prevent readministration of these vectors [50], thus causing limitations in using this type of gene therapeutic strategy. Tolerogenic rapamycin nanoparticles were used to induce immunosuppression to overcome the anti-AAV antibody issue, but they were not efficient in removing these antibodies [51]. The gram-positive *Streptococcus pyogenes* bacterium is a common human pathogen, which secretes IgG-degrading cysteine endopeptidase (called IdeS or imlifidase), which cleaves circulating IgG with a unique specificity [52,53] and is therefore useful in the treatment of several diseases [54,55,56]. Recently, IdeS treatment was shown to overcome the issue of pre-existing anti-AAV neutralizing antibodies, thus further enabling gene therapy [57].

### 2.5. Purified Inactivated Zika Vaccines (PIZV) 

The inactivated virus vaccine is produced by killing the original live virus through heat or chemicals and then introducing the remaining virus shell into the host body. Different inactivation strategies are used, such as UV, formalin and iodonaphthyl azide. It was reported that 50 and 100 µM of iodonaphthyl azide could completely inactivate ZIKV [58]. A number of PIZV candidates have been evaluated in animal models (Table 5).

Two-dose immunization of alum/adjuvant-added, purified, inactivated ZIKV vaccine (PIZV) protected AG129 mice against lethal ZIKV challenge [59]. In addition, passive immunization of naïve mice with anti-ZIKV-immune serum showed a strong positive correlation between neutralization antibody titers and protection against lethal challenge with ZIKV [59]. PIZV with aluminum hydroxide, developed by the Walter Reed Army Institute of Research and further optimized by Sanofi Pasteur, induced robust neutralizing antibody responses and provided absolute protection from challenge with a homologous ZIKV strain in immunocompetent BALB/c mice and in Cynomolgus macaques [60,61]. Two-dose vaccination of PIZV at varying concentrations, ranging from 0.016 µg to 10 µg, elicited a dose-dependent and long-lasting neutralizing antibody response in Indian rhesus macaques [62]. Complete protection against ZIKV infection was achieved with the higher PIZV doses of 0.4 µg, 2 µg, and 10 µg at 6 weeks and with 10 µg PIZV at 1-year post-vaccination [62]. Two doses of PIZV gave robust protection against ZIKV challenge in rhesus monkeys at 1-year post-vaccination [63]. A single-dose vaccination of PIZV in a dengue virus (DENV)-experienced human induced potent cross-neutralizing antibodies to both ZIKV and DENV [65]. The safety and immunogenicity of a PIZV vaccine candidate was evaluated in a double-blind, randomized, placebo-controlled phase I trial and shown to be safe and well tolerated in humans up to 52 weeks of follow-up. However, for the induction of PIZV immunogenicity, two doses were needed and, thus, not found to be durable [64]. An infectious cDNA clone of the clinical trial purified inactivated vaccine (PIV) strain PRVABC59 containing three viral replication-enhancing mutations (NS1 K265E, prM H83R, and NS3 S356F) produced more than 25-fold ZIKV than the wild type on Vero cells [66], suggesting that the cDNA clone-based manufacture platform has the advantage of higher virus yield, shortened manufacture time, and minimized chance for contamination. 

### 2.6. Virus-Like Particle (VLP)-Based Vaccines against ZIKV Infection

Virus-like particles (VLPs) are noninfectious because they are empty shell structures having no viral genome. Many viral structural proteins have the ability to assemble automatically into VLPs. They can be produced in a variety of expression systems, such as suspension cultures of mammalian, yeast and insect cells. A number of VLP-based ZIKV vaccines have been tested in animal models (Table 6).

VLPs were successfully produced by co-expression of the ZIKV structural proteins C-prM-E together with a truncated form of the protease NS3Pro linked to its cofactor NS2B constituting the viral NS2B/NS3Pro protease complex [67]. Recent studies have demonstrated that co-expression of ZIKV C-prM-E and ZIKV NS2B/NS3 [67] or WNV NS2B/NS3 protease [68] has facilitated the cleavage of C and prM, thereby allowing more efficient production of VLPs. Negative staining studies revealed that both VLPs and real virus particles are similar in size, morphology, as well as surface appearance. Indeed, the major surface glycoprotein E of ZIKV is present on the VLP surface [67], and E protein is the major target for neutralizing antibodies [68]. A VLP-based ZIKV vaccine composed of the prM gene of ZIKV located downstream of the heterogenic signal sequence and the E protein gene was generated in transiently transfected HEK293 cells [69]. A passive transfer experiment was carried out in AG129 mice, and the results showed that the VLP candidate induced a robust protective antibody response [69]. ZIKV VLPs consisting of prM and E could be quickly and easily generated by a baculovirus-insect expression system, and the VLPs stimulated high levels of ZIKV-specific neutralizing antibody titers and strong T-cell responses in all immunized mice [70]. A VLP carrier based on the hepatitis B core antigen (HBcAg) displaying the ZIKV E DIII (HBcAg-zDIII) domain was produced from *Nicotiana benthamiana* plants. Two doses of this VLP administration elicited potent humoral and cellular immune responses, which correlated with protective immunity against multiple strains of ZIKV in C57BL/6 mice [71].

VLPs mimic the conformation of natural viruses by expressing one or more structural proteins and thus stimulate robust antibodies in vivo. Antigens are present in their native conformation, but VLPs, to their advantage, use no replication virus [32]. However, application of VLP-based vaccine candidates for clinical use needs further studies.

### 2.7. mRNA-Based Vaccines against ZIKV Infection

Synthetic messenger RNAs (mRNAs) have emerged as a versatile and highly effective vaccine platform for encoding viral antigens, and they are quite attractive because of their production rapidity and flexibility. Some mRNA-based vaccines against ZIKV infection have been developed and tested in animal models (Table 6). A self-replicating mRNA vaccine encoding the ZIKV prM-E was generated, and intradermal electroporation of as little as 1 µg of this vaccine elicited potent humoral and cellular immune responses in both BALB/c and IFNAR1^−/−^ C57BL/6 mice [72]. In the latter group of mice, it resulted in complete protection from ZIKV infection [72]. To avoid ADE, a modified prM-E mRNA vaccine was designed by encoding mutations, which destroyed the conserved fusion-loop epitope in the E protein. This variant conferred protective immunity in immunocompetent mice against ZIKV infection and diminished ADE, both in vitro and in vivo [73]. Intradermal immunization of a single dose with lipid-nanoparticle-encapsulated nucleoside-modified mRNA (mRNA-LNP) encoding the prM-E proteins of a strain from ZIKV outbreak in 2013 induced potent and durable protective responses in BALB/c and C57BL/6 mice, as well as in rhesus macaques (*Macaca mulatta*) [74]. Similarly, a lipid nanoparticle-encapsulated, modified mRNA vaccine encoding ZIKV prM and E genes showed protection against placental damage and fetal death [9]. The nucleoside-modified mRNA vaccine represents a novel and promising vaccine candidate in the fight against ZIKV infection.

**Table 6 vaccines-09-01004-t006:** VLP- and mRNA-based vaccines against ZIKV infection.

Vaccine’s Name or Component	Immunogenicity in the Induction of Immune Responses	Animal Model	Vaccine Doses	Administration Route	Virus Challenged	Ref.
prM and E (HEK293 expression system)	Induced a protective antibody response	AG129 mice	Two doses at day 0 and 32	i.m.	Prototype Zika Nica 2-16 strain	[69]
prM and E (Baculovirus expression system)	Stimulated ZIKV-specific IgG and neutralizing antibodies, as well as T-cell responses	BALB/c mice	Three doses at two-week intervals	i.m.	ZIKV strain SZ-WIV01	[70]
EDIII (*Nicotiana benthamiana* plant expression system)	Elicited potent humoral and cellular immune responses correlated with protective immunity against multiple strains	C57BL/6 mice	Three doses at three-week intervals	s.c.	Puerto Rico strain PRVABC59	[71]
prM and E	Intradermal electroporation of as little as 1 µg of this vaccine elicited potent humoral and cellular immune responses in BALB/c and IFNAR^−/−^ C57BL/6 mice, resulting in complete protection of the latter mice against ZIKV infection.	BALB/c and IFNAR^−/−^ C57BL/6 mice	Two doses at four-week interval	i.d.	ZIKV strain MR-766	[72]
	Conferred protection and sterilizing immunity in immunocompetent mice against ZIKV infection and diminished ADE in vitro, as well as in vivo	AG129, BALB/c and C57BL/6 mice	Single dose and two doses at three-week interval	i.m.	African ZIKV strain (Dakar 41519)	[73]
	Induced potent and durable protective responses in mice and non-human primates	BALB/c and C57BL/6 mice; rhesus macaques (*Macaca mulatta*)	Single dose	i.d.	Puerto Rico strain PRVABC59	[74]

Note: i.m., intramuscular injection; s.c., subcutaneous injection; i.d., intradermal injection.

### 2.8. Other Types of Vaccines against ZIKV Infection

Apart from the above-mentioned vaccine platforms, other types of vaccine candidates are also available against ZIKV. A chimeric ZIKV with DENV-2 prM-E genes was highly potent in A129 mice, and it induced robust neutralizing antibody responses, which conferred complete protection from challenge with ZIKV [75]. Over the last several years, computer-assisted peptide vaccines have started to draw much attention as alternative vaccine candidates. The peptide vaccine is a more focused approach to precisely locate the epitope region within the antigens and elicit immune responses. Occasionally, multiple peptides for one or more viral infections can be combined into clusters for immunization against a wide spectrum of infections [76].

The potential advantages and disadvantages of the above mentioned different types of vaccines are summarized in Table 7.

## 3. Animal Models as Tools to Assist in the Development of Zika Vaccines 

To develop vaccines against ZIKV infection, establishment and development of specific animal models is critically important. Animal models can be developed in a number of ways, using the effects of host genetic status and immune function upon ZIKV infection [43]. Until now, scientists around the world have established murine and NHP models to study the infection mechanisms of ZIKV on the central nervous system, to test the immunogenicity of the vaccine candidates, and to assess their protective efficacy against viral infection [23,44,63,77], as well as deepen our understanding of the causes of microcephaly, which, in turn, has contributed to our understanding of ZIKV and the protective immunity of hosts. The NHP model is useful to derive correlates of protection for vaccine studies, but it does not recapitulate all the clinical signs observed in humans [78]. AG129 mice, which lack both IFN-α/β and -γ receptors, but elicit B-cell and T-cell responses to infection, support ZIKV replication and high virus load in organs exhibiting severe disease symptoms with progression to mortality [79,80]. As type I interferon signaling in both B- and CD4^+^ T-cells is required for optimal antibody response to virus infection, vaccine studies in this mouse model do not provide a full measure of immune correlates of protection [78,81]. Nevertheless, it is an effective animal model to study vaccine efficacy against viremia, disease pathogenesis, and mortality.

## 4. Conclusions

Currently, more efforts are needed to prevent and/or treat ZIKV infection. The development of a safe and efficacious ZIKV vaccine remains a global health priority. Vaccine development and use on a large scale to prevent pandemics involve several factors such as mass production of vaccine, adjuvant selection, establishment of optimal animal models for preclinical studies, validation of safety and efficacy in animal models and clinical trials in different parts of the world using a large cohort of patients, immunization strategies, storage conditions, as well as manufacturing and production costs. Novel proposals for multi-epitope vaccines, as well as the discovery of new adjuvant formulations and delivery systems that could enhance and/or modulate immune responses, may help pave the way for development of successful vaccine candidates.

## Figures and Tables

**Figure 1 vaccines-09-01004-f001:**
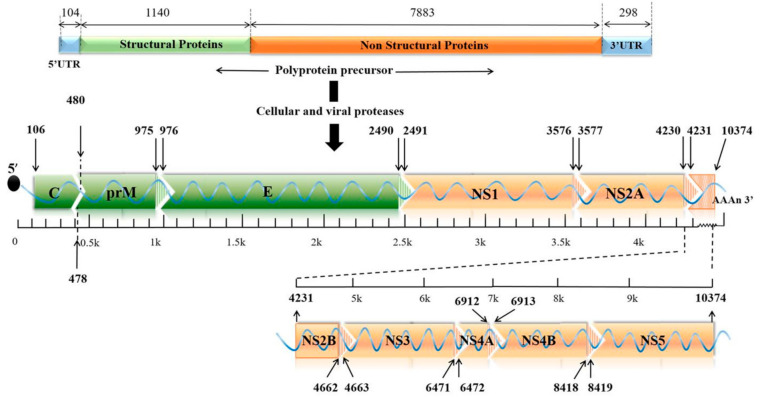
The genomic structure of ZIKV. The genome contains a single ORF encoding three structural proteins (C, prM, and E) and seven non-structural proteins (NS1, NS2A, NS2B, NS3, NS4A, NS4B, and NS5) with two UTRs at both ends. ORF: opening reading frame; C: capsid; prM: premembrane; E: envelope; UTR: untranslated region.

**Figure 2 vaccines-09-01004-f002:**
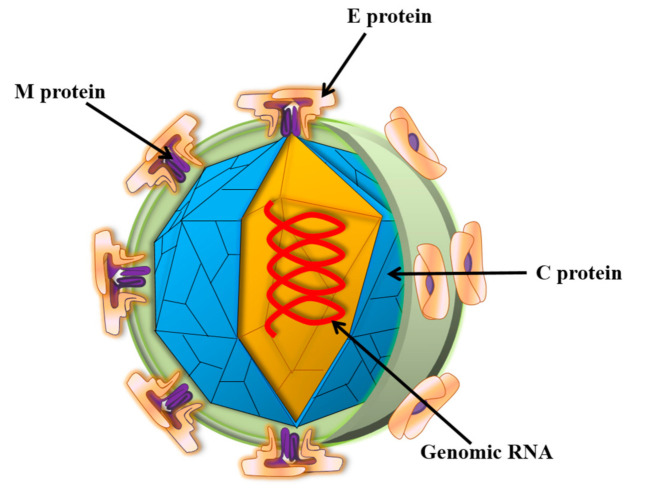
Schematic diagram of ZIKV structure. Zika virions are enveloped, spherical, and approximately 50 nm in diameter. The surface prM/M and E proteins are symmetrically arranged on the viral surface. M: membrane; E: envelope; C: capsid.

**Table 1 vaccines-09-01004-t001:** DNA vaccines against ZIKV infection.

Vaccine’s Name or Component	Immunogenicity in the Induction of Immune Responses	Animal Model	Vaccine Doses	Administration Route	Virus Challenged	Ref.
prM and E	Completely protected mice against ZIKV-associated damage to the testes and sperm and prevented viral persistence in the testes	Type-I interferon knockout mice	Two doses at two-week interval	i.m.	Puerto Rico Strain PRVABC59	[17]
pVAX1-ZME (prM and E)	Induced robust ZIKV-specific cellular and long-term humoral immune responses with high and sustained neutralizing activity, which provided passive protection against ZIKV infection in neonatal mice	BALB/c mice	Three doses at three-week intervals	i.m.	(SMGC-1 strain, GenBank accession number: KX266255	[18]
GLS-5700 (prM and E)	Prevented fertility loss in male IFNAR^−/−^ mice	C57BL/6J mice and IFNAR^−/−^ mice	Two doses at two-week interval	i.m.	Puerto Rico Strain PRVABC59	[19]
VRC5288 and VRC5283	Induced detectable T-cell response and antibody response with neutralization activity. The immunogenicity of VRC5283 was better than that of VRC5288.	Humans	Single dose, two and three doses	i.m	No	[20]
prM and E	Elicited protective responses against multiple diverse ZIKV isolates	C57BL/6c mice	Four doses at days 0, 24, 42, and 199	i.m	Puerto Rico Strain PRVABC59	[21]
pVAX-NS1, pVAX-tpaNS1, pVAX-tpaNS1-IMX313P (NS1)	pVAX-tpaNS1 vaccination induced significantly higher NS1-specific antibody titers and CD4^+^, as well as CD8^+^, T-cell responses compared to pVAX-NS1 and pVAX-tpaNS1- IMX313P	BALB/c and IFNAR^−/−^ mice	Three doses at two-week intervals	i.d.	ZIKVzkv2015	[22]

Note: prM, anterior membrane; E, envelope; IFNAR^−/−^, type I IFN receptor–deficient; i.m., intramuscular injection; i.d., intradermal injection.

**Table 2 vaccines-09-01004-t002:** Subunit vaccines against ZIKV infection.

Vaccine’s Name or Component	Immunogenicity in the Induction of Immune Responses	Animal Model	Vaccine Doses	Administration Route	Virus Challenged	Ref.
E	Induced robust antigen binding IgG titers and high levels of neutralizing antibodies in the mice, which protected against viremia after ZIKV infection	Swiss Webster, BALB/c, and C57BL/6 mice	Three doses at three-week intervals	i.m.	Puerto Rico Strain PRVABC59	[26]
	Induced high neutralizing antibody titers	Cynomolgus macaques and BALB/c mice	Three doses at three-week intervals	i.m.	Puerto Rico Strain PRVABC59	[27]
EDIII	Induced high titer of IgG and ZIKV-neutralizing antibodies and showed no evidence of ADE induction in mouse serum	C57BL/6 mice	Four doses at three-week intervals	s.c.	Puerto Rico Strain PRVABC59	[28]
E90 (Consisting of the first 450 amino acids at the N-terminal region of E protein)	Immunization of pregnant mice with E90 protected the developing brains of offspring, both in utero and in the neonatal period, from subsequent ZIKV infection and microcephaly. E90 induced robust ZIKV-specific humoral responses in adult BALB/c mice.	ICR (CD-1 immunocompetent) mice; BALB/c mice	Two doses at two-week interval	i.p.	GZ01 and FSS13025 strains	[29,30]
EDIII fragments (E296–406; E298–409; E301–404)	Induced sustained broad-spectrum neutralization antibodies and passive transfer of the E298–409-specific antibodies prevented ZIKV infection in newborns and immunocompromised adults.	BALB/c mice and A129 mice	Five doses at days 0, 21, 42, 210, and 300	i.m.	R103451 and FLR strains	[31]

Note: i.m., intramuscular injection; s.c., subcutaneous injection; i.p., intraperitoneal injection.

**Table 3 vaccines-09-01004-t003:** Live-attenuated vaccines against ZIKV infection.

Vaccine’s Name or Component	Immunogenicity in the Induction of Immune Responses	Animal Model	Vaccine Doses	Administration Route	Virus Challenged	Ref.
ZIKV-3′ UTR-10-LAV	Showed complete protection from viremia and induced a saturated neutralizing antibody response	A129 mice	Single dose	s.c.	Cambodian strain FSS13025 and Puerto Rico strain PRVABC59	[34]
ZIKV-3′ UTR-20-LAV	Induced strong immune responses and protected ZIKV-induced damage to testes in mice; induced sterilizing immunity in NHPs	A129 mice and rhesus macaques	Single dose	s.c.	Cambodian strain FSS13025 and Puerto Rico strain PRVABC59	[35]
ZIKV-NS1-LAV (NS1)	Markedly diminished viral RNA levels in maternal, placental, and fetal tissues, which resulted in protection against placental damage and fetal death	A129 mice andrhesus macaques	Single dose	s.c.	Puerto Rico strain PRVABC59	[36]
LAV (with 9-amino-acid deletion in the C protein)	Not only elicited protective immunity that completely prevented viremia, morbidity and mortality, but also fully prevented infection of pregnant mice and maternal-to-fetal transmission	A129 mice	Single dose	s.c.	Puerto Rico strain PRVABC59	[37]

Note: s.c., subcutaneous injection.

**Table 4 vaccines-09-01004-t004:** Virus-vector-based vaccines against ZIKV infection.

Vaccine’s Name or Component	Immunogenicity in the Induction of Immune Responses	Animal Model	Vaccine Doses	Administration Route	Virus Challenged	Ref.
Ad4-prM-E and Ad5-prM-E	Ad5-prM-E vaccination induced both humoral and T-cell responses, while Ad4-prM-E induced only a T-cell response.	C57BL/6 mice	Two doses at three-week interval	i.m.	Puerto Rico strain PRVABC59	[39]
hAd5-prM-E	Induced both cell-mediated and humoral immune responses, which conferred protection against a ZIKV challenge	C57BL/6 mice and *Ifnar1*^−/−^ mice	Single dose	i.n.	Puerto Rico strain PRVABC59	[41]
Ad5-Sig-prM-Env (prM-E) and Ad5-Env (E)	Both vaccines elicited robust humoral and cellular immune responses in immunocompetent BALB/c mice, as well as in A129 mice, but Ad5-Sig-prM-Env-vaccinated mice resulted in significantly higher ZIKV-specific neutralizing antibody titers and lower viral loads than Ad5-Env-vaccinated mice.	BALB/c mice and A129 mice	Single dose	i.m.	Puerto Rico strain PRVABC59	[42]
ChAdOx1	Induced high levels of protective responses in challenged mice	BALB/c mice	Single dose	i.m.	Brazilian ZIKV	[43]
RhAd52-prMEnv	Induced ZIKV-specific neutralizing antibodies in rhesus monkeys; antibodies sufficient for protection against ZIKV challenge in mice	Rhesus monkeys and BALB/c mice	Single dose	i.m.	Brazilian ZIKV and Puerto Rico strain PRVABC59	[44]
rVSV-prM-E-NS1	Induced ZIKV-specific antibody and T-cell immune responses that conferred partial protection against ZIKV infection	A129 mice and BALB/c mice	Single dose	i.n.	Cambodian strain FSS13025	[45]
VSV-Capsid and VSV-ZikaE260-425	Both vaccines induced strong ZIKV-specific humoral responses in immunized BALB/c mice, but VSV-Capsid immunization elicited significantly higher levels of IFN-γ+ CD8^+^ and CD4^+^ T-cells than that of VSV-ZikaE260-425 vaccine.	BALB/c mice	Single dose	i.n.	Puerto Rico strain PRVABC59	[46]

Note: i.m., intramuscular injection; i.n., intranasal injection.

**Table 5 vaccines-09-01004-t005:** Inactivated vaccines against ZIKV infection.

Vaccine’s Name or Component	Immunogenicity in the Induction of Immune responses	Animal Model	Vaccine Doses	Administration Route	Virus Challenged	Ref.
Alum-adjuvant mixed purified inactivated ZIKV vaccine (PIZV)	Two-dose vaccination of the candidates was highly immunogenic in the mouse models, which protected AG129 mice against lethal ZIKV challenge. Passive transfer of naïve mice with ZIKV-immune serum also showed full protection against lethal ZIKV challenge.	CD-1 and AG129 mice	Three doses at four-week intervals	i.m.	Puerto Rico strain PRVABC59	[59]
	Induced robust neutralizing antibody responses and provided complete protection from homologous ZIKV strain challenge	BALB/c mice and cynomolgus macaques	Two doses at three/four-week interval	i.m.	Puerto Rico strain PRVABC59	[60,61]
PIZV	Elicited a dose-dependent and long-lasting neutralizing antibody responses	Indian rhesus macaques	Two doses at four-week interval	i.m.	Puerto Rico strain PRVABC59	[62]
	Two dose-vaccination of the Type of vaccine gave a robust protection against ZIKV challenge.	rhesus macaques	Two doses at four-week interval	s.c., i.m.	Brazil ZKV2015	[63]
	Safe and well tolerated in humans up to 52 weeks of follow-up; but two doses not durable for immunogenicity required	Phase I clinical trial	Single dose and two doses at two/four-week interval	i.m.	No	[64]

Note: i.m., intramuscular injection; s.c., subcutaneous injection.

**Table 7 vaccines-09-01004-t007:** Potential advantages and disadvantages of different types of Zika vaccines.

Vaccine Types	Advantages	Disadvantages	Ref.
DNA vaccines	Chemically stable and cost effective; easy and safe to scale up; can induce both humoral and cellular immune responses and are capable of mediating long-term protection	Have the potential of integrating the exogenous gene into the host genome, leading to induction of host autoimmunity	[17,18,19,20,21,22,25]
Subunit vaccines	Rapid, stable, and consistent production	Normally need multiple doses with appropriate adjuvants	[26,27,28,29,30,31]
Live-attenuated vaccines	Single dose could induce high immune responses, rapid induction of durable immunity	Safety problems and need cold-chain storage facilities	[24,34,35,36,37]
Virus-vector-based vaccines	Single dose could induce higher and faster immune responses with lasting protection	Pre-existing immunity problem	[39,41,42,43,44,45,48,49]
Inactivated vaccines	Easy production and storage; convenient to make multivalent vaccines	Safety problems; need multiple injections; unable to deal with mutant viruses	[59,60,61,62,63,64]
VLP-based vaccines	Noninfectious and could induce robust antibodies; multiple choices of expression systems	Application for clinical use needs further studies	[69,70,71]
mRNA-based vaccines	Rapid and flexible production; could induce potent humoral and cellular immune responses	Need cold-chain storage facilities; new technology, lack of historical accumulation	[72,73,74]

## Data Availability

Not applicable.

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
