# Peer review of "Current Progress in the Development of Zika Virus Vaccines"

_vaccines, 2021, doi:10.3390/vaccines9091004_

Round 1
Reviewer 1 Report
This review article drafted by Kehui Zhou et al. covers most of the advances in Zika virus vaccine development. This review article has appropriately cited all the research articles in the field and summarized the findings. Below are the reviewer’s comments.
It will be useful if the authors would have generated a table to compare and contrast different types of vaccines. Though it would be a common feature of the vaccines and can be found elsewhere, still generating a concise table in this review will be valuable to the reader.
All the tables also indicate the dose of vaccines, virus challenges, route of vaccine administration using a separate column for each.
Reviewer 2 Report
This is a well-structured and useful review wit up to date references. I have a few minor comments and questions for the authors to address:
- Abstract line 1 'occult' is an unusual term- insidious might be better
- Abstract line 45 -keep to present tense, so 'discussed' should be 'discuss'
- line 41 Introduction- please reference the statement about common epitopes between Dengue and Zika -is it reference[10]?
- Line 59 -present tense, therefore 'summarise' and 'discuss'
- Line122 -what is CoVaccine HT? also 10mg seems like a very high dose?
- line 180- 'focus-forming units' seems an unusual term; should this be plaque-forming units?
- line 185-what is the route of delivery with the novel Tri-Grid?
- Table 4: what is meant by 'suitable levels' of protective response?
